# The Autophagy Regulatory Molecule CSRP3 Interacts with LC3 and Protects Against Muscular Dystrophy

**DOI:** 10.3390/ijms21030749

**Published:** 2020-01-23

**Authors:** Can Cui, Shunshun Han, Shuyue Tang, Haorong He, Xiaoxu Shen, Jing Zhao, Yuqi Chen, Yuanhang Wei, Yan Wang, Qing Zhu, Diyan Li, Huadong Yin

**Affiliations:** Farm Animal Genetic Resources Exploration and Innovation Key Laboratory of Sichuan Province, Sichuan Agricultural University, Chengdu 611130, China; cuican123@stu.sicau.edu.cn (C.C.); hanshunshun@stu.sicau.edu.cn (S.H.); tangshuyue@stu.sicau.edu.cn (S.T.); hehaorong@stu.sicau.edu.cn (H.H.); shenxiaoxu@stu.sicau.edu.cn (X.S.); zhaojing@stu.sicau.edu.cn (J.Z.); chenyuqi@stu.sicau.edu.cn (Y.C.); zhuqing@sicau.edu.cn (Q.Z.); diyanli@sicau.edu.cn (D.L.)

**Keywords:** CSRP3, myoblasts, autophagy, atrophy, apoptosis

## Abstract

CSRP3/MLP (cysteine-rich protein 3/muscle Lim protein), a member of the cysteine-rich protein family, is a muscle-specific LIM-only factor specifically expressed in skeletal muscle. CSRP3 is critical in maintaining the structure and function of normal muscle. To investigate the mechanism of disease in CSRP3 myopathy, we performed siRNA-mediated CSRP3 knockdown in chicken primary myoblasts. CSRP3 silencing resulted in the down-regulation of the expression of myogenic genes and the up-regulation of atrophy-related gene expressions. We found that CSRP3 interacted with LC3 protein to promote the formation of autophagosomes during autophagy. CSRP3-silencing impaired myoblast autophagy, as evidenced by inhibited autophagy-related ATG5 and ATG7 mRNA expression levels, and inhibited LC3II and Beclin-1 protein accumulation. In addition, impaired autophagy in CSRP3-silenced cells resulted in increased sensitivity to apoptosis cell death. CSRP3-silenced cells also showed increased caspase-3 and caspase-9 cleavage. Moreover, apoptosis induced by CSRP3 silencing was alleviated after autophagy activation. Together, these results indicate that CSRP3 promotes the correct formation of autophagosomes through its interaction with LC3 protein, which has an important role in skeletal muscle remodeling and maintenance.

## 1. Introduction

Skeletal muscle development and physiology are regulated by many factors and mechanisms. In recent years, evidence has suggested that CSRP3 plays a vital role in skeletal muscle physiology [1,2]. CSRP3 belongs to the cysteine and glycine-rich protein family of proteins comprising CSRP1, CSRP2, and CSRP3. CSRP3, a cytoskeletal protein, is richly expressed in skeletal muscle and cardiac muscle and interacts with telethonin, α-actinin, calcineurin, and cofilin-2, with functions in muscle development and maintenance [3,4,5]. CSRP3 expression levels and intracellular localization are associated with many skeletal myopathies, such as nemaline myopathy, facioscapulohumeral muscular dystrophy (FSHD), and limb-girdle muscular dystrophy type 2B [6,7]. Although CSRP3 has been reported to be involved in muscle mass and myopathy, the cellular and molecular mechanisms underlying the function of muscle LIM protein (MLP) in regulating muscle myopathy remain unclear.

Autophagy is a critical cellular process to achieve the metabolic needs of cells and renewal of organelles and maintain the cells themselves in a steady-state [8]. During autophagy, cytosolic proteins, and damaged organelles are encapsulated by bilayer membrane vesicles, forming autophagosomes, which are then transported into lysosomes for degradation. The precise balance of autophagy plays an important role in maintaining skeletal muscle mass and muscle health [9]. Excessive autophagy can lead to rapid loss of muscle mass because of the continued removal of essential cellular components, whereas inadequate autophagy leads to chronic loss of muscle mass because of the accumulation of damaged or aging cellular components [10,11].

The LIM domain is a zinc finger structure, and several types of proteins, including homologous domain transcription factors, kinases, and proteins, contain one or more LIM domain [12]. Proteins containing LIM domains play an important role in biological processes such as cytoskeletal tissue, organ development, and tumor formation [13,14,15]. The LIM domain is involved in protein–protein interactions [16]. Recently, several reports have demonstrated that LIM-domain proteins may play a major role in regulating cell differentiation by interacting with autophagic pathway factors, such as FHL1 [17], FHL2 [18], and PDLim1 [19]. Moreover, Rashid et al. found that CSRP3 at the intersection of autophagy and mechanotransduction in mice, but there is no relation in chicken [20].

In this study, we investigated the function of CSRP3 in chicken skeletal muscle development in vivo and in vitro and explored the regulation of muscular atrophy disease by CSRP3 through affecting autophagy.

## 2. Results

### 2.1. CSRP3 Silencing Results in Skeletal Muscle Atrophy

To examine the effect of CSRP3 on skeletal muscle development, chicks were injected with lentivirus expressing siRNA targeting CSRP3 for CSRP3-knockdown (CSRP3-KO); controls (control-knockdown, Ctrl-KO) were established in parallel. We confirmed that CSRP3-KO in chick breast muscle resulted in significantly decreased CSRP3 mRNA and protein levels (Figure 1A,B). H&E staining revealed a general decrease in myofiber size and the presence of necrotic myofibers in the CSRP3-KO group (Figure 1C). The muscle weight and muscle fiber cross-sectional area in the CSRP3-KO group were significantly lower than those in controls (Figure 1D,E). We next investigated the effect of CSRP3-KO on atrophy-related genes in skeletal muscle and found that CSRP3-KO significantly increased Atrogin-1 and MuRF-1 mRNA levels and Atrogin-1 protein expression (Figure 1F,G). These data suggested that CSRP3 participates in regulating muscular atrophy of skeletal muscle.

### 2.2. CSRP3 Regulates the Differentiation of Skeletal Muscle Cells

To further determine the effect of CSRP3 on the development of chicken skeletal muscle, we silenced CSRP3 in chicken primary myoblasts to investigate the effect on myoblast differentiation (Figure 2A,B). We found that CSRP3 siRNA decreased the mRNA levels of three myogenic marker genes including myogenin (MyoG), myoglobin (Mb), and myosin heavy chain (MyHC) and the protein level of MyoG and MyHC compared with controls (Figure 2C,D). In addition, we found that CSRP3 silencing resulted in significantly increased expression of atrophy-related genes including Atrogin-1 and MuRF-1 (Figure 2E,F). These data suggested that CSRP3 is involved in regulating the development of chicken skeletal muscle.

### 2.3. Gene Expression Analysis of CSRP3 Silenced Cells

To further study the molecular regulation mechanisms of CSRP3 during muscle development, we performed transcriptome sequencing analysis on si-Ctrl and si-CSRP3 cells after 3 days of culture in differentiation medium. In CSRP3 knockdown cells, we detected significantly decreased expression of a set of genes that are involved in myogenesis, including myosin heavy chain 1 (Myh1), myoglobin (Mb), troponin T type 1 (Tnnt1), and creatine kinase muscle (Ckm), which are important genes that regulate muscle development (Figure 3A). Gene ontology analysis revealed that these genes were significantly enriched in the functional annotations of myogenesis and muscle cell development (Figure 3B). Notably, we also found that differentially expressed genes were also enriched in the autophagy pathway (Figure 3C). Therefore, we speculated that CSRP3 may regulate skeletal muscle development through autophagy.

### 2.4. CSRP3 Silencing Results in Reduced Autophagy

We next examined the potential function of CSRP3 in autophagy by knockdown of CSRP3 in chicken myoblasts. Cells transfected with CSRP3-siRNA showed significantly decreased expression of ATG5 and ATG7 mRNA (Figure 4A). Then, we monitored the autophagy flux by examination of endogenous LC3, P62, and Beclin-1, and all of them were decreased in CSRP3-silenced cells compared with controls (Figure 4B). Immunofluorescence showed that the numbers of LC3 small punta were significantly reduced after CSRP3 was silenced, and no ring-shaped structures were observed in CSRP3-silenced cells (Figure 4C,D). These results indicated that CSRP3 regulates autophagy flux in muscle cells.

### 2.5. CSRP3 Interacts with LC3 to Regulate Autophagosome Formation

To further study the mechanism of CSRP3 in regulating autophagy in chicken myoblasts, we performed immunofluorescence and co-immunoprecipitation assays to assess whether CSRP3 co-localized with the autophagosome marker LC3 in myoblasts. Immunofluorescence showed that the distribution of CSRP3 and LC3 proteins in myoblasts was similar, while the expression of CSRP3 and LC3 decreased after CSRP3-silencing, which indicated they were co-expressed (Figure 5A). Co-immunoprecipitation further showed that CSRP3 and LC3 interact in cells, and decreased co-immunoprecipitation was found in CSRP3-silenced myoblasts compared with control cells (Figure 5B). These results suggested that CSRP3 interacts with LC3 to promote the formation of autophagosomes.

### 2.6. CSRP3 Protects Against Apoptosis by Regulating Autophagy

Previous studies have shown that autophagy is also associated with apoptosis. We thus used the autophagy inhibitor 3-methyl adenine (3-MA) and induced rapamycin (RAP) to inhibit and promote autophagy in cells, respectively. We found that the numbers of apoptotic bodies increased significantly after adding 3MA, but the number of apoptotic bodies in the control group and the RAP group was very small (Figure 6A). Flow cytometry demonstrated that the percentage of apoptotic cells increased significantly in CSRP3-silenced cells compared with controls (Figure 6B). In addition, CSRP3 silencing significantly increased the levels of cleaved caspase-3 and caspase-9 protein expression (Figure 6C). We next further investigated the regulatory mechanism of CSRP3 in apoptosis. We found that apoptosis of cells with CSRP3 silencing was further aggravated with 3MA treatment (Figure 6D,E). Notably, the apoptosis rate of CSRP3 silenced cells was reduced after incubation with RAP (Figure 6F,G). These results suggested that apoptosis induced by silencing of CSRP3 may be due to impaired autophagy.

## 3. Discussion

Recent studies have demonstrated that CSRP3 is an important regulator in the physiology and pathophysiology of striated muscle [21]. CSRP3 mutation can directly lead to human myocardial and skeletal muscle diseases [22]. In addition, some evidence has shown that the CSRP3 and MyoD interaction plays an important role in myogenic differentiation and remodeling of muscle cells [23]. In the present study, we found that CSRP3 promotes myogenesis in vitro and in vivo. Buyandelger et al. reported that CSRP3 is a cytoskeleton-associated protein that acts as a stress sensor in the muscle [24]. Ilona et al. also found that CSRP3 plays structural and functional roles in skeletal muscle and that CSRP3 knockout is associated with cardiomyopathy and heart failure [25]. Here we showed that when CSRP3 is knocked down, the myogenic gene expression is down-regulated and the muscular dystrophy protein expression is up-regulated. These data suggest that CSRP3 plays a crucial regulatory function in muscle development and maintenance.

Autophagy plays an important function in skeletal muscle maintenance by regulating protein homeostasis, and impaired autophagy disrupts the fitness and integrity of skeletal muscles [26]. We found that CSRP3 regulates basal autophagy and exerts a beneficial role in chicken primary myoblasts. Beclin-1 is involved in autophagic vesicle nucleation via its interaction with Vps34, a Class III phosphatidylinositol 3-kinase [27]. We found that CSRP3 silencing decreased the expression levels of LC3II and Beclin-1. Therefore, the silencing of CSRP3 may lead to autophagy deficiency by inhibiting the expression of LC3 and Beclin-1. In addition, a previous study showed that the LIM domain of CSRP3 interacts with LC3 to facilitate the autophagy process [20]. Our colocalization and coimmunoprecipitation results were consistent with this finding. A previous study showed that the complex working mode of CSRP3 located in the nucleus and cytoplasm can regulate skeletal muscle development through different action modes [23]. In chronic obstructive pulmonary disease, the persistence of dysfunctional organelles and cytoplasmic proteins after autophagy inhibition is important for the activation of catabolic pathways [28]. In mice with ATG5 and ATG7 knockout, muscle loss, protein aggregates, and accumulation of abnormal membranous structures were observed [29,30,31]. Therefore, various myopathies caused by mutations of CSRP3, such as dilated cardiomyopathy, heart failure and muscular dystrophy, may be caused by insufficient autophagy.

Like autophagy, apoptosis plays a vital role in cell development [32]. Previous studies showed that several proteins operate in both autophagy and apoptotic pathways, insufficient autophagy contributes to the accumulation of waste material inside the cell and the induction of apoptosis, however, and the increase of autophagy is considered a protective mechanism against apoptosis [33]. For example, in nutrient-deficient or growth factor-deficient cells, increased autophagy keeps cells alive by protecting them from apoptosis [34]. In addition, apoptosis is elevated in ATG5-deficient mice, leading to cell death [35]. Elliott et al. reported that autophagy is necessary for myoblast differentiation to prevent apoptosis [36]. Here we showed that CSRP3 knockdown resulted in the induction of apoptosis in chicken primary myoblasts. However, in cells silenced for CSRP3, apoptosis was reduced after the addition of the autophagy inducer RAP, indicating that the increased apoptosis of CSRP3 silenced cells is regulated by the autophagy pathway. These results suggested that inhibition autophagy resulting in apoptosis and suppression of protein synthesis, which leads to muscle fibrosis and muscle weakness.

The autophagic, apoptotic, and necrotic cell death pathways play important roles not only in skeletal muscle but also in cardiac tissues [37,38,39]. Impaired autophagy can lead to the premature death of cardiomyopathy and myocardium [40]. Furthermore, these cell death pathways contribute to myocardial function and sudden cardiac death, including ventricular arrhythmias and ventricular fibrillation. Ullrich et al. reported that lack of SPRED2 impinges on autophagy, leading to cardiac dysfunction and life-threatening arrhythmias [41]. Meyer et al. found that autophagy was increased during reperfusion in fibrillated mouse hearts [42]. Studies have reported that inhibiting caspases during early reperfusion protects the myocardium from fatal reperfusion injury [43,44]. These findings suggest that increased autophagy during myocardial perfusion may inhibit caspase activity, thereby protecting cardiomyocytes from damage. CSRP3 also plays an important role in the development and regulation of myocardium. CSRP3-deficient mice are characterized by the destruction of myocardial cell structure, dilated cardiomyopathy, and heart failure [22]. One potential reason for this pathological phenomenon may be that CSRP3-deficiency inhibits autophagy in cardiomyocytes, resulting in myocardial damage.

In summary, our study shows that CSRP3 plays a vital role in the activation of autophagy through its interaction with LC3 protein, and deficiency of CSRP3 leads to apoptosis in chicken myoblasts. Because autophagy is involved in multiple pathological processes in skeletal muscle, a better understanding of the molecular regulatory mechanisms of CSRP3 in autophagy will be important for the treatment of muscle disease.

## 4. Materials and Methods

### 4.1. Animal Procedures

The specialized broiler chickens (ROSS 308) were purchased from the Xinjin Yunda Poultry Breeding Cooperative (Chengdu, Sichuan Province, China). Housing, breeding, and all experimental protocols used in this study strictly followed with the guidelines of the Animal Welfare Committee of the Faculty of Agriculture at the Sichuan Agriculture University, and the approval number is 201810200602 (2018-10-20).

### 4.2. Lentiviral Intramuscular Injections

Lentiviral intramuscular injection was followed by the previous description of Luo et al. (2018) [45] with some modification. Briefly, 1-day-old chicks were infected with lentiviruses (Hanbio, Shanghai, China) at the dosage of 1× 10^6^ IU/mL by direct injection into the breast muscle.

### 4.3. Cell Cultures

Chicken primary myoblasts were isolated from 12-day old embryonic chickens (Ross 308, a standard broiler breed) according to a previous study [46]. Cells were expanded in growth medium (GM) with Dulbecco’s Modified Eagle Medium (DMEM; Sigma, MO, USA), 15% fetal bovine serum (FBS, Gibco, Grand Island, NY, USA), and 1% penicillin-streptomycin (Solarbio, Beijing, China). After primary myoblasts achieved 90% confluence, the GM was replaced by differentiation medium (DM) with DMEM (Sigma, St. Louis, MO, USA), 2% horse serum (Hyclone, Logan, UT, USA), 1% penicillin-streptomycin (Solarbio, Beijing, China) to induce differentiation. Cell culture reagents were purchased from TransGen Biotech (Beijing, China), unless otherwise specified.

### 4.4. CSRP3 Knockdown

Myoblasts were placed in 6-well plates and cultured approximately 80% of the cells in each well were fused and transfected with CSRP3 siRNA. CSRP3 siRNA sequences: (Sense: GCAGCUCACGAAUCUGAAATT; Antisense: UUUCAGAUUCGUGAGCUGCTT, provided by Sangon Biotech (Shanghai, China). Cells were transfected with Lipofectamin^TM^ 3000 Transfection Reagent according to the instructions of the manufacturer (Invitrogen, Carlsbad, CA, USA). The cells were collected at differentiation medium 3 d post-transfection for the extraction of RNA and protein to detect knockdown efficiency.

### 4.5. Autophagy Treatment

To block or induce autophagy, cells were transfected CSRP3 siRNA or Ctrl siRNA for 72 h, and then incubated with 3-methyladenine (10 mM; sigma) or rapamycin (5 μm; Selleckchem, Houston, TX, USA) for 6 h.

### 4.6. RNA Extraction and Real-Time PCR (RT-PCR)

Total RNA from breast muscle and myoblasts were extracted using TRIzol reagent (Invitrogen, Carlsbad, CA, USA) according to the instructions of the manufacturer. Oligo (dT) primers were used to produce the first strand of cDNA using M-MLV Reverse Transcriptase (ThermoFisher, Waltham, MA, USA) for PCR analysis. RT-PCR analysis was performed by the CFX96-Touch^TM^ real-time PCR detection system (Bio-Rad, Hercules, CA, USA). The RT-PCR primers were as Table 1. The gene expression was normalized to β-actin following the 2^−ΔΔCT^ method [47].

### 4.7. Western Blot and Immunoprecipitation Analysis

After removing the culture medium, cells were washed with PBS three times, and the RIPA lysis (Sigma) buffer was used for lysis. Protein samples (200μg) were separated by 12% SDS-polyacrylamide gel electrophoresis and then transferred to a PVDF membrane (Beyotime). The PVDF membrane was incubated with 5% defatted milk powder for 1 h at room temperature and then incubated at 4 °C overnight with specific primary antibodies. The PVDF membrane was rinsed with TBST (Beyotime, Shanghai, China) and stained with HRP-labeled secondary antibody for 1 h at room temperature. After washing by TBST (Beyotime), the bands were visualized with ECL reagent (Amersham Pharmacia Biotech, Piscataway, NJ, USA).

For immunoprecipitation analysis, the cells were lysed with IP lysis buffer (Beyotime), and lysates were immunoprecipitated with LC3 or CSRP3 antibodies. Immunocomplexes were washed with IP lysis buffer three times and then analyzed by western blot. Images were acquired from an EU-88 image scanner (GE Healthcare, King of Prussia, PA, USA), and quantification of protein bands was determined using the Quantity One 1-D software (version 4.4.0; Bio-Rad, Hercules, CA, USA).

The primary antibody used in this experiment including CSRP3 (Santa Cruz Biotechnology Inc., Heidelberg, Germany); MyHC (Santa); LC3B (Sigma, St. Louis MO, USA); MyoG (Biorbyt, Cambridge, UK); Atrogin-1 (Abcam, Cambridge, UK); Beclin-1 (Cell Signaling Technology, Mass, USA); caspase-3 (Abcam); caspase-9 (Absin, Beijing, China); β-actin (Beyotime, Beijing, China); the second antibody were used including in this experiment: goat anti-rabbit HRP (Zen Bioscience, Chengdu, China); goat anti-mouse HRP (Zen Bioscience), TRITC-Goat anti-Rabbit IgG (Beyotime), FITC-Goat anti-Mouse IgG (Beyotime).

### 4.8. Histological and Morphometric Analysis

Chicken breast muscle tissues were obtained from groups infected with siRNA: pLKO.1-CSRP3 and pLKO-control, and the number was 15 in every case. According to the study of Luo et al. [45], we performed an initial dose on 9 d. Tissues were quickly harvested and then fixed in 10% formalin. After processing, the fixed tissues were stained with hematoxylin and eosin (H&E) and then subjected to image analysis.

### 4.9. Immunofluorescence and Confocal Microscopy

Myoblasts were cultured in 12-well plates and fixed in 4% formaldehyde for 30 min after transfection for 48 h. Cells were washed with PBS three times and permeabilized with 0.5% Triton X-100 (Sigma) for 10 min. Next, the cells were blocked with goat serum (Gibco, Grand Island, NY, USA) for 1 h and then incubated with primary antibody in PBS-1% goat serum (Gibco) at 4 °C overnight. The cells were then incubated with fluorescent secondary antibody in the dark for 90 min. The cell nuclei were visualized by DAPI staining (Sigma). A fluorescence microscope (Olympus, Melville, NY, USA) was used to visualize cells.

### 4.10. Apoptosis Analysis

Cells cultured in 12-well plates were transfected with CSRP siRNA for 48 h when they achieved 50% confluence. Cells were then collected and fixed in 70% ethanol at 4 °C overnight. The fixed cells were stained using 50 μg/mL propidium iodide solution (Sigma) containing 10 μg/mL RNase A and 0.2% (*v*/*v*) Triton X-100 in the dark at room temperature for 30 min. Flow cytometry analysis was performed on a flow cytometer (BD Pharmigen, San Diego, CA, USA), and data were processed using FlowJo7.6 software (Tree star, San Carlos, CA, USA).

### 4.11. RNA Sequencing

The total RNA was extracted from CSRP3 siRNA and Ctrl siRNA according to the above experimental steps. The cDNA library construction, sequencing, and transcriptome data analysis were performed by Beijing Biomarker Technologies Co., Ltd. (Beijing, China).

### 4.12. Statistical Analysis

Statistical analyses were performed using SPSS 17.0 software (SPSS Inc., Chicago, IL, USA). Data are shown as least squares means ± standard error of the mean (SEM). *p* < 0.05 was considered statistically significant.

## Figures and Tables

**Figure 1 ijms-21-00749-f001:**
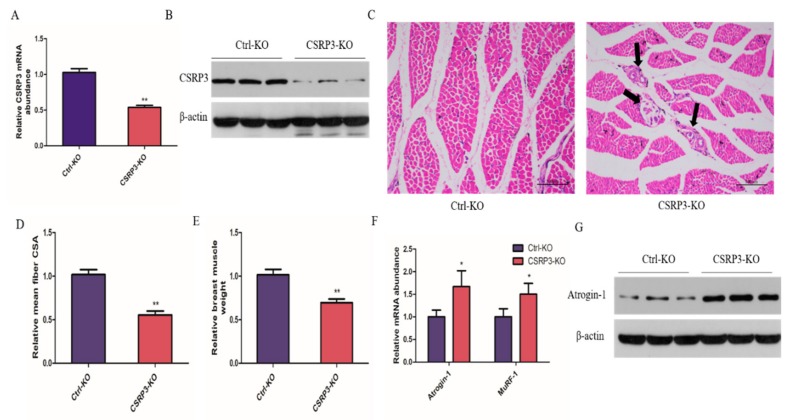
CSRP3 regulates mean muscle fiber and muscle mass in chicken skeletal muscles. (**A** and **B**) CSRP3 mRNA and protein expression in breast muscle after lentivirus injected. Hematoxylin and eosin (H&E) staining (**C**), mean muscle fiber CSA, scale bar = 100 μm (Black arrows point to necrotic myofibers) (**D**), and muscle weight (**E**) of breast muscle fiber cross-section after lentivirus injected. (**F** and **G**) Atrogin-1 and MuRF-1 mRNA expression and Atrogin-1 protein expression in breast muscle after lentivirus injected. Data are expressed as mean ± SEM of three independent experiments. * *p* < 0.05, ** *p* < 0.01.

**Figure 2 ijms-21-00749-f002:**
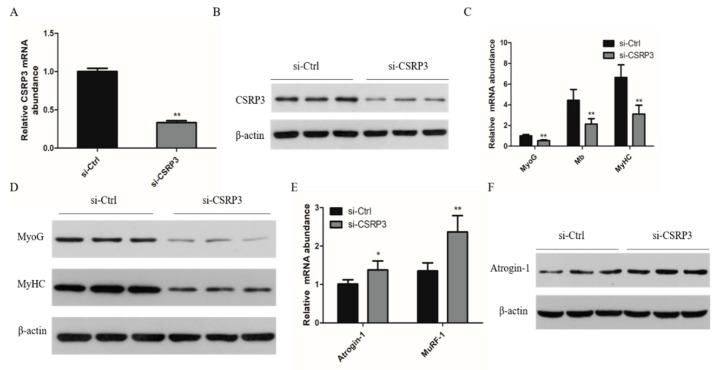
CSRP3 regulates the development of skeletal muscle myoblasts in chicken. (**A** and **B**) CSRP3 mRNA and protein in myoblasts transfected with si-CSRP3 of si-Ctrl. (**C** and **D**) The mRNA expression of MyoG, Mb, and MyHC, and protein abundance in myoblasts transfected with si-CSRP3 of si-Ctrl. (**E** and **F**) The mRAN and protein level of Atrogin-1 and MuRF-1 mRNA in myoblasts transfected with si-CSRP3 of si-Ctrl. Data are expressed as mean ± SEM of three independent experiments. * *p* < 0.05, ** *p* < 0.01.

**Figure 3 ijms-21-00749-f003:**
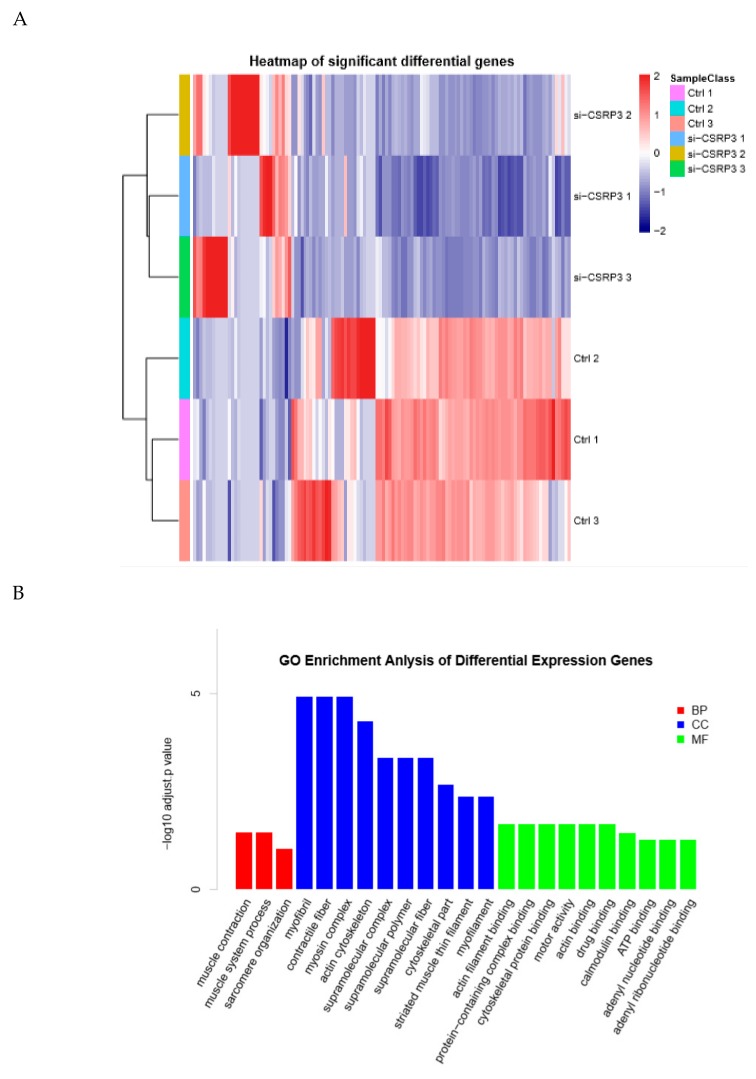
Gene expression analysis in ctrl and CSRP3 silenced cells by RNA sequencing. (**A**) Hierarchical clustering and heatmap of significant difference genes between chicken myoblasts transfected with control and CSRP3 siRNA. (**B**) Gene ontology (GO) enrichment analysis of significant differential expression genes between chicken myoblasts transfected with control and CSRP3 siRNA. (**C**) Pathway enrichment analysis of significant differential expression genes between chicken myoblasts transfected with control and CSRP3 siRNA.

**Figure 4 ijms-21-00749-f004:**
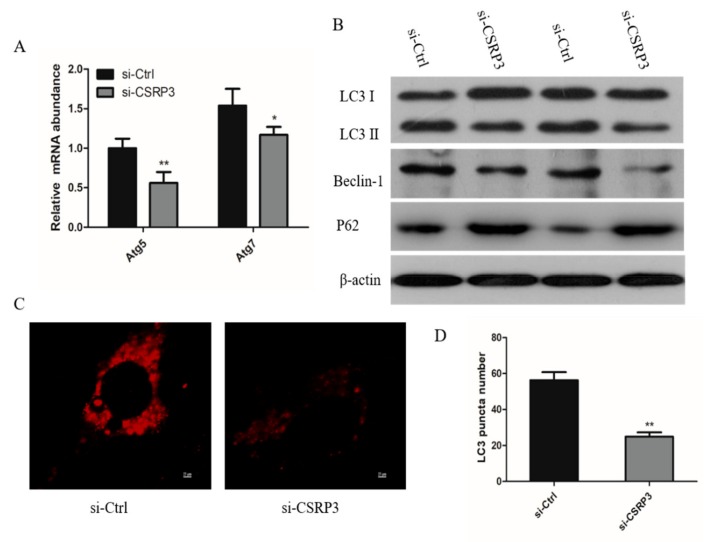
CSRP3 regulates autophagy in chicken myoblasts. (**A**) ATG5 and ATG7 mRNA expression in ctrl or CSRP3 siRNA cells. (**B**) Western blot analysis revealed LC3, P62, and Beclin-1 protein levels in ctrl and CSRP3 siRNA cells. (**C**) Immunofluorescence analysis of cell transfected with ctrl or CSRP3 siRNA then stained with LC3; scale bar = 25 μm. (**D**) The right diagram shows the average number of LC3 puncta in ctrl or CSRP3 siRNA cells. Data are expressed as mean ± SEM of three independent experiments. * *p* < 0.05, ** *p* < 0.01.

**Figure 5 ijms-21-00749-f005:**
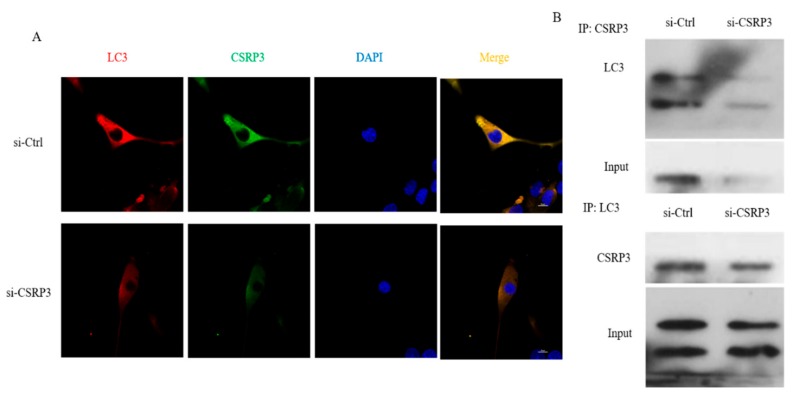
Co-localization of CSRP3 and LC3 in chicken myoblasts. (**A**) Representative confocal microscopy images of LC3 (red) and CSRP3 (green) were shown; scale bar = 20 μm. (**B**) Reciprocal co-immunoprecipitation analysis between CSRP3 and LC3 in ctrl and CSRP3 siRNA cells. Data are expressed as mean ± SEM of three independent experiments.

**Figure 6 ijms-21-00749-f006:**
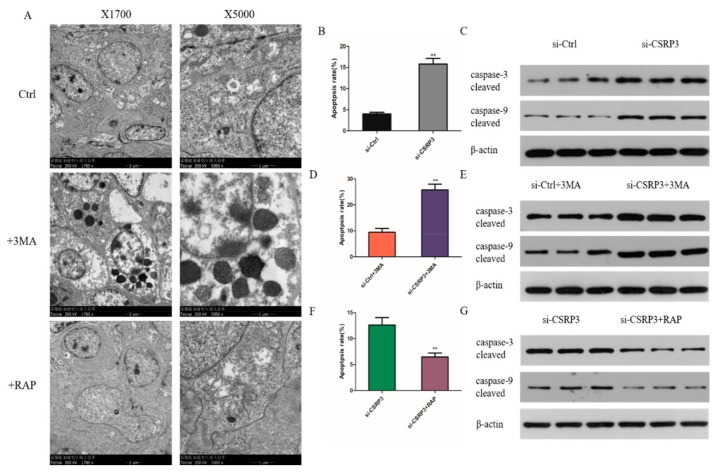
CSRP3 silence promotes cell apoptosis. (**A**) The ultrastructure of cells treated with 3MA, RAP or DMSO (ctrl) was observed by transmission electron microscopy. (**B**) Flow cytometry showed the apoptosis rates in control and CSRP3 siRNA cells. (**C**) The cleaved of caspase-3 and caspase-9 protein levels in control and CSRP3 siRNA cells. (**D**) Flow cytometry showed the apoptosis rates in control and CSRP3 siRNA cells incubated with 3MA. (**E**) Western blot analysis cleaved caspase-3 and caspase-9 protein levels in controls and CSRP3 siRNA cells incubated with 3MA. (**F**) Flow cytometry showed the apoptosis rates in control and CSRP3 siRNA cells untreated or treated with rapamycin (RAP). (**G**) The cleaved of caspase-3 and caspase-9 protein levels in CSRP3 siRNA cells untreated or treated with RAP. Data are expressed as mean ± SEM of three independent experiments.

**Table 1 ijms-21-00749-t001:** Gene-special primers for RT-PCR.

Gene	Forward Primer (5′-3′)	Reverse Primer (5′-3′)
CSRP3	CCCTCCACACCAACTAACCC	TCTGCAGCGTACACCGATTT
MyoG	CGGAGGCTGAAGAAGGTGAA	CGGTCCTCTGCCTGGTCAT
Mb	CCCTGAGACTTTGGATCGCTT	CTGGGATTTTGTGCTTCGTGG
MyHC	CTCCTCACGCTTTGGTAA	TGATAGTCGTATGGGTTGGT
Atrogin-1	TCAACGGGTCGGCAAGTCT	TCCCTCCCATCGCTCAGTC
MuRF-1	GGCAGCAGCATCATCTCGG	CCTCGCAGGTGACGCAGTAG
Atg5	GATGAAATAACTGAAAGGGAAGC	TGAAGATCAAAGAGCAAACCAA
Atg7	TCAGATTCAAGCACTTCAGA	GAGGAGATACAACCACAGAG
β-actin	CCGCTCTATGAAGGCTACGC	CTCTCGGCTGTGGTGGTGAA

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
