# Peer review of "The Autophagy Regulatory Molecule CSRP3 Interacts with LC3 and Protects Against Muscular Dystrophy"

_ijms, 2020, doi:10.3390/ijms21030749_

Round 1
Reviewer 1 Report
Commentary sheet to the manuscript: Autophagy regulatory molecule, CSRP3, interacts with LC3 and protects muscular dystrophy
The research topic of the submitted manuscript is very interesting and relevant to the journal. In recent years a lot of evidence suggests that regulation of autophagy and apoptosis play a key role in skeletal muscle physiology. Although, recently more data has been reported on CSRP3 important role in development of muscle mass, the cellular and molecular mechanisms underlying its function in regulating muscle myopathy remain unclear. I found the work is an interesting novelty. However, due to the manuscript has many shortcomings and mistakes major revision is required before publishing it in a high-quality journal.
Comments
In general, the manuscript to be overall well written and well organized. On the other hand, I found that some very important points are inadequate or completely missing. Therefore, I suggest you to be careful and do thorough work while doing major revision of your work.
Introduction
In this part the information about CSRP3 role and about the general role of autophagy is too long, while the purpose of the research is not clearly stated. I advise you to pay more attention to the specific purpose of your research rather than writing more lines of unnecessarily information.
Results
The results reported here supported by a wide range of experiments. I am convinced that the authors performed careful and thorough experimental work.
Materials and Methods
The description of the methods is not always complete, some very important elements are missing, therefore please complete the missing items with necessary information:
Animal ProceduresIt is not mentioned what kind of animals were used. Please, write in more detail the experimental protocol for animal procedures.
Cell CulturesThe DM 3d term is not exactly clear for me. Please, remember all the methods must be clearly written so as anybody could repeat the experimental design!
Treatment protocols and antibodiesI advise you do not combine these two parts. My suggestion is to split this part into two different chapters: a) Treatment protocols and b) Antibodies. Please, write the treatment protocols more precisely!
Also, I must mention that in the Results you wrote about the Gene expression analysis in control and CSRP3 silenced cells. However, in the Materials and Methods section the description of this method is not even mentioned! Please, add this missing method to the appropriate section!
Figures
Although the figures present nice data and cover useful information, they are not always of the best quality.
Figure 1:
The pictures representing the western blot results are not properly captioned. It is not mentioned which band represents the given group! The names of the CSRP3 and CSRP3-KO groups must be exactly indicated! The figure explanation under the Figure 1 has a mistake: in line 74 instead of (D) you must write (G) (picture of Western blot analysis of Atrofin-1 protein level).
Figure 2:
The mistakes mentioned above are true for this figure as well. Please, insert the names of the groups for all the western blot pictures, make it clear which is the Control and which is the siRNA group.
Figure 3:
The figure B is not of the best quality! It is missing the proper labelling. I understand that to generate good co-immunoprecipitation data sometimes is a big challenge, however it is advisable to improve the quality of this figure.
Finally, the autophagic, apoptotic, and necrotic cell deaths also play important roles in mammals, not only in the skeletal muscle, but even in cardiac tissues (apoptotic, autophagic, and necrotic cell deaths). In addition, these pathological cell deaths contribute to myocardial function and sudden cardiac death, including ventricular arrhythmias and ventricular fibrillation. Therefore, the following papers should be discussed in the Discussion Section, in a separated paragraph, of the revised version: Br J Pharmacol. 2000 May;130(2):197-200. PMID: 10807653, DOI: 10.1038/sj.bjp.0703336; Naunyn Schmiedebergs Arch Pharmacol. 2001 Dec;364(6):501-7, PMID: 11770004, DOI: 10.1007/s002100100483; Curr Pharm Des, 2013, PMID: 23590156, DOI: 10.2174/138161281939131127122510; J Cell Mol Med. 2017 Jun;21(6):1058-1072. doi: 10.1111/jcmm.13053. Epub 2016 Dec 20. Review; Int J Mol Sci. 2019 Apr 2;20(7). pii: E1628. doi: 10.3390/ijms20071628, Autophagy. 2019 Dec 29:1-16. doi: 10.1080/15548627.2019.1704117; J Mol Cell Cardiol. 2019 Apr;129:13-26. doi: 10.1016/j.yjmcc.2019.01.023. Epub 2019 Feb 13.
Author Response
Response to Reviewer 1 Comments
Dear reviewer:
Thank you very much for your comments concerning our manuscript entitled “Autophagy regulatory molecule, CSRP3, interacts with LC3 and protects muscular dystrophy” (ID:IJMS-688918). Those comments are all valuable and very helpful for revising and improving our paper, as well as the important guiding significance to our researches. We have studied comments carefully and have made correction. I hope this revision can make my paper more acceptable. The main corrections in the paper and the responds to your comments are addressed point by point.
Point 1: In this part the information about CSRP3 role and about the general role of autophagy is too long, while the purpose of the research is not clearly stated. I advise you to pay more attention to the specific purpose of your research rather than writing more lines of unnecessarily information.
Response 1: We thank the reviewer’s suggestion, we have added some sentences in the section of introduction, such as line 44-48, 52-53.
Point 2. It is not mentioned what kind of animals were used. Please, write in more detail the experimental protocol for animal procedures.
Response 2: We thank the reviewer’s suggestion. We have described the experimental material and the animal procedures in detail, such as “4.1 Animal Procedures “and “4.2 Lentiviral intramuscular injections”.
Point 3. The DM 3d term is not exactly clear for me. Please, remember all the methods must be clearly written so as anybody could repeat the experimental design!
Response 3: We thank the reviewer’s suggestion, it means that 3 days in the differentiation medium, and the manuscript has been revised.
Point 4. L I advise you do not combine these two parts. My suggestion is to split this part into two different chapters: a) Treatment protocols and b) Antibodies. Please, write the treatment protocols more precisely!
Response 4: We thank the reviewer’s suggestion, the manuscript has been revised.
Point 5. Also, I must mention that in the Results you wrote about the Gene expression analysis in control and CSRP3 silenced cells. However, in the Materials and Methods section the description of this method is not even mentioned! Please, add this missing method to the appropriate section!
Response 5: We are very sorry that I have omitted this part, and we added the “4.11 RNA sequencing” in the Materials and Methods section.
Point 6. The pictures representing the western blot results are not properly captioned. It is not mentioned which band represents the given group! The names of the CSRP3 and CSRP3-KO groups must be exactly indicated! The figure explanation under the Figure 1 has a mistake: in line 74 instead of (D) you must write (G) (picture of Western blot analysis of Atrofin-1 protein level).
Response 6: We thank the reviewer for pointing out these inconsistencies, and we have revised them in the manuscript.
Point 7. The mistakes mentioned above are true for this figure as well. Please, insert the names of the groups for all the western blot pictures, make it clear which is the Control and which is the siRNA group.
Response 7: We thank the reviewer for pointing out these inconsistencies, and we have revised them in the manuscript.
Point 8. The figure B is not of the best quality! It is missing the proper labelling. I understand that to generate good co-immunoprecipitation data sometimes is a big challenge, however it is advisable to improve the quality of this figure.
Response 8: Based on the reviewer’s comment, we have replaced it with a high quality picture.
Point 9. Finally, the autophagic, apoptotic, and necrotic cell deaths also play important roles in mammals, not only in the skeletal muscle, but even in cardiac tissues (apoptotic, autophagic, and necrotic cell deaths). In addition, these pathological cell deaths contribute to myocardial function and sudden cardiac death, including ventricular arrhythmias and ventricular fibrillation. Therefore, the following papers should be discussed in the Discussion Section, in a separated paragraph, of the revised version: Br J Pharmacol. 2000 May;130(2):197-200. PMID: 10807653, DOI: 10.1038/sj.bjp.0703336; Naunyn Schmiedebergs Arch Pharmacol. 2001 Dec;364(6):501-7, PMID: 11770004, DOI: 10.1007/s002100100483; Curr Pharm Des, 2013, PMID: 23590156, DOI: 10.2174/138161281939131127122510; J Cell Mol Med. 2017 Jun;21(6):1058-1072. doi: 10.1111/jcmm.13053. Epub 2016 Dec 20. Review; Int J Mol Sci. 2019 Apr 2;20(7). pii: E1628. doi: 10.3390/ijms20071628, Autophagy. 2019 Dec 29:1-16. doi: 10.1080/15548627.2019.1704117; J Mol Cell Cardiol. 2019 Apr;129:13-26. doi: 10.1016/j.yjmcc.2019.01.023. Epub 2019 Feb 13.
Response 9: Thank you for your suggestion. We have made the more detailed discussion on the autophagy in cardiac tissues. Per the reviewer’s comment, we have added a separated paragraph in the discussion part from line 205-218.
We tried our best to improve the manuscript and made some changes in the manuscript. These changes will not influence the content and framework of the paper. We appreciate for Reviewer’s warm work earnestly, and hope that the response and correction will meet with approval.
Once again, thank you very much for your comments and suggestions
Reviewer 2 Report
The manuscript is of interest but, in my opinion, requires revisions.
Major concerns:
The paragraph 2.3 could be ameliorated with a more detailed description of the results. In the paragraph 2.4 the Authors monitored autophagic flux in si-CRSP3 and they concluded that there was a reduction of LC3-II and Beclin-1. Since SQSTM1/p62 is a good indicator of autophagic flux, data regarding the expression of this classical autophagy marker could be added to support the results. The paragraph 2.6 should be rephrased. It is difficult for the reader to follow the results. Furthermore, since cross talk between autophagy and apoptosis is a crucial point in the manuscript, double-label immunofluorescence experiments with LC3 antibodies in conjunction with antibodies directed to cleaved caspase-3 could be helpful to support the results obtained after the treatment with 3MA or RAP. Figures should be ameliorated: the different experimental groups should be indicated in western blotting images (Fig 1B, 1G; 2B, 2D, 2F) considering that the Authors describe two different experimental conditions “control or CSRP3-KO” (Fig 1) and “control and si-CRSP3” (Fig 2); a scale bar should be added to microscope images (Fig 1C; 4C; 5A); Fig 4C should be ameliorated, more representative images could be used to show differences between control and si-CRSP3. The Discussion section could be ameliorated. Since cross talk between autophagy and apoptosis is a major issue, this important point needs to be further supported. The emphasis Authors claim on this aspect is not adequately addressed.Author Response
Response to Reviewer 2 Comments
Dear reviewer:
Thank you very much for your comments concerning our manuscript entitled “Autophagy regulatory molecule, CSRP3, interacts with LC3 and protects muscular dystrophy” (ID:IJMS-688918). Those comments are all valuable and very helpful for revising and improving our paper, as well as the important guiding significance to our researches. We have studied comments carefully and have made correction. I hope this revision can make my paper more acceptable. The main corrections in the paper and the responds to your comments are addressed point by point.
Point 1: The paragraph 2.3 could be ameliorated with a more detailed description of the results.
Response 1: We thank the reviewer’s suggestion, more detailed description have been made in the manuscript.
Point 2. In the paragraph 2.4 the Authors monitored autophagic flux in si-CRSP3 and they concluded that there was a reduction of LC3-II and Beclin-1. Since SQSTM1/p62 is a good indicator of autophagic flux, data regarding the expression of this classical autophagy marker could be added to support the results.
Response 2: We thank the reviewer’s suggestion, we supplemented the expression of P62 protein after transfected with CSRP3 siRNA and control siRNA in Figure 4B.
Point 3. The paragraph 2.6 should be rephrased. It is difficult for the reader to follow the results.
Response 3: Per the reviewer’s comment, we have rewritten this paragraph.
Point 4. Furthermore, since cross talk between autophagy and apoptosis is a crucial point in the manuscript, double-label immunofluorescence experiments with LC3 antibodies in conjunction with antibodies directed to cleaved caspase-3 could be helpful to support the results obtained after the treatment with 3MA or RAP.
Response 4: We thank the reviewer’s suggestion. Due to species limitations, we did not find a caspase-3 antibody that could perform immunofluorescence on chickens, but we observed the ultrastructure of cells through electron microscopy to explain the effect of 3MA and RAP treatment on apoptosis. And fluorescence tests on the activity of caspase-3 after treatment with 3 MA have been investigated by MCMILLAN, as explained in our discussion. (McMillan, Elliott M., and Joe Quadrilatero. "Autophagy is required and protects against apoptosis during myoblast differentiation." Biochemical Journal 462.2 (2014): 267-277.)
Point 5. Figures should be ameliorated: the different experimental groups should be indicated in western blotting images (Fig 1B, 1G; 2B, 2D, 2F) considering that the Authors describe two different experimental conditions “control or CSRP3-KO” (Fig 1) and “control and si-CRSP3” (Fig 2) a scale bar should be added to microscope images (Fig 1C; 4C; 5A);
Response 5: We thank the reviewer’s suggestion, we have made corrections in the manuscript.
Point 6. Fig 4C should be ameliorated, more representative images could be used to show differences between control and si-CRSP3.
Response 6: We thank the reviewer’s suggestion, we have taken more representative images.
Point 7. The Discussion section could be ameliorated. Since cross talk between autophagy and apoptosis is a major issue, this important point needs to be further supported. The emphasis Authors claim on this aspect is not adequately addressed.
Response 7: We thank the reviewer’s suggestion, and we have added some more sentences to discuss the interaction between autophagy and apoptosis in the manuscript. In addition, according to another reviewer’s comment, we also added more detailed discussion on the autophagy in cardiac tissues in a separated paragraph.
We tried our best to improve the manuscript and made some changes in the manuscript. These changes will not influence the content and framework of the paper. We appreciate for Reviewer’s warm work earnestly, and hope that the response and correction will meet with approval.
Once again, thank you very much for your comments and suggestions
Round 2
Reviewer 1 Report
The authors made a substantial revision. The manuscript is very clear right now. This reviewer has no additional commet. The manuscript should be accepted for publication in the Journal.